# Collagen VI Is a Gi-Biased Ligand of the Adhesion GPCR GPR126/ADGRG6

**DOI:** 10.3390/cells12111551

**Published:** 2023-06-05

**Authors:** Caroline Wilde, Paulomi Mehta Chaudhry, Rong Luo, Kay-Uwe Simon, Xianhua Piao, Ines Liebscher

**Affiliations:** 1Rudolf Schönheimer Institute of Biochemistry, University of Leipzig, 04103 Leipzig, Germany; 2Department of Pediatrics, Boston Children’s Hospital, Boston, MA 02467, USA; 3Newborn Brain Research Institute, University of California, San Francisco, CA 94158, USA; 4Weill Institute for Neuroscience, University of California, San Francisco, CA 94158, USA; 5Eli and Edythe Broad Center of Regeneration Medicine and Stem Cell Research, University of California, San Francisco, CA 94158, USA

**Keywords:** adhesion GPCR, GPR126, ADGRG6, biased signaling, extracellular matrix ligand, collagen VI

## Abstract

GPR126/ADGRG6, a member of the adhesion G-protein-coupled receptor family, balances cell differentiation and proliferation through fine-tuning of intracellular cAMP levels, which is achieved through coupling to Gs and Gi proteins. While GPR126-mediated cAMP increase has been proven to be essential for differentiation of Schwann cells, adipocytes and osteoblasts, Gi-signaling of the receptor was found to propagate breast cancer cell proliferation. Extracellular ligands or mechanical forces can modulate GPR126 activity but require an intact encrypted agonist sequence, coined the *Stachel*. Even though coupling to Gi can be seen for constitutively active truncated receptor versions of GPR126 as well as with a peptide agonist derived from the *Stachel* sequence, all known N-terminal modulators have so far only been shown to modulate Gs coupling. Here, we identified collagen VI as the first extracellular matrix ligand of GPR126 that induces Gi signaling at the receptor, which shows that N-terminal binding partners can mediate selective G protein signaling cascades that are masked by fully active truncated receptor variants.

## 1. Introduction

GPR126/ADGRG6 belongs to the adhesion G-protein-coupled receptor (aGPCR) family [1,2,3,4]. Like most aGPCRs, GPR126 is defined by a very large extracellular membrane region (ECR), which harbors five structurally distinguishable domains: Complement C1r/C1s, Uegf, Bmp1 (CUB), Pentraxin (Pent), Hormone Receptor (HormR) and GPCR Autoproteolysis INducing (GAIN), as well as a Sperm protein, Enterokinase and Agrin (SEA) domain [5] (Figure 1A). The receptor undergoes autoproteolytic cleavage [6,7] at the GAIN domain [8], resulting in the formation of an N-terminal fragment (NTF) and C-terminal fragment (CTF) that remain non-covalently associated at the cell membrane. GPR126 carries an encrypted agonistic sequence coined the *Stachel*, which is located at the very N-terminal portion of the CTF. GPR126 activation can be achieved through mutational unmasking of the encrypted agonist or *Stachel*-sequence-derived peptides [3]. This basal, peptide or mutation-induced activation usually results in an increase in cAMP levels, which indicates Gs-coupling [3]. Using a chimeric Gqi protein, however, downstream IP3 accumulation was also observed, indicating the possibility for Gi coupling even though Gs coupling appears to be the dominant activation pathway through the *Stachel* sequence [2]. GPR126 plays many key roles in development and disease. Through an increase in cAMP levels, GPR126 contributes essentially to myelination, development of the peripheral nervous system in zebrafish, mouse and human (PNS) [1,2,9,10] and differentiation of osteoblasts [11], as well as adipocytes [12]. A role for potential Gi coupling remained elusive until a recent study indicated that Gi signaling of GPR126 promotes proliferation of breast cancer cells, which was induced through progesterone or 17-hydroxyprogesterone [13] (Figure 1B).

In line with the stated activation pattern of GPR126, its extracellular allosteric ligands, collagen IV [14], as well as Prion protein PrPC [15], or application of mechanical forces such as shaking or vibration [16], only activate Gs-mediated signaling (Figure 1B). Even the observed cAMP reduction through incubation with laminin 211 under static conditions was merely an inhibition of the Gs-protein-signaling cascade [16], while the addition of mechanical forces reversed the inhibition, allowing for cAMP elevation.

Here, we establish collagen VI as a biased Gi-coupling-inducing ECM ligand of GPR126, which lays ground to the assumption that an intermediate activation state can exist in the full-length receptor that is distinct from the full activation mode induced by NTF removal.

## 2. Materials and Methods

### 2.1. Materials

If not stated otherwise, all standard substances were purchased from Sigma Aldrich (Taufkirchen, Germany), Merck (Darmstadt, Germany), and C. Roth GmbH (Karlsruhe, Germany). Cell culture material was obtained from ThermoFisher Scientific (Schwerte, Germany). Peptide synthesis was carried out using Core Unit Peptide Technology (Medical Faculty, Leipzig University, Leipzig, Germany). Peptide solution from purified powder was achieved by preparing a 100 mM in 100% DMSO solution, which was further diluted into 10 mM stocks using a 50 mM, pH 8 Tris buffer and finally pH-controlled. Peptide concentration used in assays was 1 mM concentration. The 1 mM peptide solution contained 1% DMSO and 10% of Tris buffer used for dilution. Empty vector transfected cells were treated accordingly.

### 2.2. Methods

#### 2.2.1. GPR126 Plasmid Construction and Fusion Protein Generation

Three different mouse GPR126 fragments were constructed into the pFUSE-hFc2 vector (Invitrogen) that contains the IL2 signal peptide sequence and the human IgG Fc tag. Mouse GPR126 N terminus and its truncated fragments were generated by PCR and subsequently fused to hFc to generate human IgG Fc-tagged GPR126N 31-807 (GPR126N 31-807-hFc), GPR126N 31-438-hFc and GPR126N 446-807-hFc by inserting into the EcoR I/Bgl II sites. The PCR primers for mouse GPR126N are as follows:Forward Primer 31: 5′-CACGAATTCGGTTCCTCTCTCAGTGTGCGG-3′;Reverse Primer 807: 5′-GTTAGATCT GCA CAG ACA AAT GGT CTC ACC-3′Reverse Primer 438: 5′-GTTAGATCT GACTTTCATCCT GTCCTC TCC-3′Forward Primer 446F: 5′-CACGAATTCG GACAAAAGG TTG GTG CTC TGC-3′

Mouse IgG Fc-biotag GPR126 constructs: the mouse pSecTag2A-mFc-Biotag vector was a gift from H. H. Lin (Chang Gung University, Taiwan). The mouse GPR126 sequence was fused into the vector as previously published [17]. Mouse IgG Fc-tagged construct was generated via PCR. Mouse GPR126N 31-438 PCR product was inserted between AscI/BamHI in vector (contain a signal peptide). The primers are:mFc 31-438 F (AscI): 5′-GCCAGGCGCGCCcctctctcagtgtgcg gatgtggcag-3′mFc 31-438 R (BamHI): 5′-CTCTGGATCCGAGACTTTCATCCTGTCCTCTTAC-3′

GPR126N 446-807 PCR product was inserted between AscI/KpnI in the vector. The primers are:mFc448-807 F (AscI): 5′-GCCAGGCGCGCCTTGGTGCTCTGGGCCCTTCTAGTC-3′mFc 448-807 R (KpnI): 5′-CTCTGGTACCTTGCACAGACAAATGGTCTCACC-3′

All expression constructs were sequenced to confirm their identity.

#### 2.2.2. Generation of mFc and hFc Fusion Protein

Each of the above GPR126N expression constructs was transiently transfected into HEK-293T cells (obtained from ATCC). The culture media was changed to serum-reduced OPTI-MEM 24 h after transfection. The conditioned media was harvested 48–72 h later, and concentrated as previously described [18]. The concentrated proteins were purified through protein A column (GE Healthcare, Chicago, IL, USA).

#### 2.2.3. In Vitro Biotinylation of mGPR126-mFc Fusion Proteins

Purified mFc fusion proteins were buffered in 10 mM Tris-HCl, pH 8.0, by dialysis and incubated with 1 μL of BirA enzyme (Avidity, Denver, CO, USA) overnight at room temperature. Excess biotin was subsequently removed by dialysis with 10 mM Tris-HCl, pH 7.3, containing 10 mMCaCl2 and 100 mM NaCl. Following the confirmation of integrity of the biotinylated proteins using Western blotting probed with Extravidin-HRP (Sigma), the proteins were quantified by dot-blot analysis using myelin basic protein–biotin (Avidity, Denver, CO, USA) as standard and stored at −80 °C.

#### 2.2.4. Purification of GPR126 Immunocomplexes and Tandem MS and Sequencing

Pull-down assay was performed as previously described [18]. Briefly, purified GPR126NTF-mFc-biotag and mFc control proteins were biotinylated in vitro with BirA enzyme. For sciatic nerve lysate, adult wild-type sciatic nerves were prepared as previously described [18]. Briefly, sciatic nerves from adult (postnatal day 45–60) C57/BL6 mice were dissected under a Leica stereo microscope (MZ 6; Leica Pte Ltd., Buffalo Grove, IL, USA), and powdered on liquid nitrogen, then lysed in ice-cold RIPA buffer (1% Nonidet P-40, 50 mM Tris, pH 7.6, 120 mM NaCl, and 1 mM EDTA) containing protease inhibitor cocktail set 1 (Calbiochem). The lysates were cleared of insoluble materials by centrifugation at 16,000× *g* for 10 min at 4 °C. The protein concentration was determined by a Bio-Rad protein assay method (Bio-Rad) according to the manufacturer’s protocol, and equal amounts of protein were used for each pull-down assay. Biotinylated GPR126-NTF—amino acid (AA) 31-348 and AA446-807—as well as mFc proteins were separately incubated with mouse sciatic nerve lysates, and immunocomplexes were affinity-purified by streptavidin beads (Sigma). GPR126-associated proteins were eluted in 2× SDS loading buffer, subjected to SDS/PAGE and Coomassie blue stain.

The whole lane was used for MS at the Taplin Biological Mass Spectrometry at Harvard Medical School. Mass spectrometry data were searched against the mouse International Protein Index (IPI mouse 339) database using the protein identification software Mascot (v2.2.04, Matrix Science, Boston, MA, USA) [18].

#### 2.2.5. Co-Immunoprecipitation (co-IP) and Western Blot Analysis

Co-immunoprecipitation experiments were performed as previously described [18], using the sciatic nerve as the collagen VI source. For sciatic nerve lysate, adult wild-type sciatic nerves were prepared as previously described [16,19]. A total of 8 µg of protein was incubated with two separate mouse GPR126-NTF fragments (NTF-31-348 and NTF-446-807) overnight at 4 °C. Protein-G beads were used to pull down the GPR126N-hFc protein complex. The entire immune complexes from each co-IP were denatured with 2× SDS sample buffer after extensive washes, separated by 10% SDS-PAGE gel and then transferred to PVDF membrane. The membrane was first blotted with rabbit anti-human IgG Fc antibody (Thermo Scientific, Waltham, MA, USA) following standard protocols. Subsequently, the membrane was stripped and re-probed with rabbit polyclonal anti-collagen VI antibody (Fitzgerald #70R-CR009x) using standard Western blot protocols.

#### 2.2.6. In Vitro Functional Assays

The human N- and C-terminally tagged GPR126 (previously described and characterized in [3,16]) was heterologously expressed in COS-7 cells grown in Dulbecco’s minimum essential medium (DMEM) supplemented with 10% fetal bovine serum (FBS), 100 units/mL penicillin and 100 μg/mL streptomycin at 37 °C and 5% CO_2_ in a humidified atmosphere. For assays, COS-7 cells were split into 96-well plates (1.5 × 10^4^ cells/well) or 12-well plates (15 × 10^4^ cells/well) and transfected with LipofectamineTM 2000 (ThermoFisher Scientific) according to the manufacturer’s protocol. A total of 50 ng (96-well plates) or 1 µg (12-well plates) of receptor plasmid DNA/well were used for cAMP accumulation assays. The same amounts of receptor DNA and 10 ng (96-well plates) or 200 ng (12-well plates) of Gqi4 chimera plasmid were used when Gi-dependent IP accumulation was determined. To perform second messenger assays, transfected cells were stimulated with the indicated concentration of human collagen VI (GTX27538, GeneTex, Irvine, CA, USA) or with agonistic peptides (pGPR126). Empty vector-transfected cells were treated with assay medium containing the same concentration of the respective ligand. To inhibit Gi proteins, cells were incubated with DMEM containing 100 ng/mL of Pertussistoxin for 16 h prior to stimulation.

For cAMP measurements, 48 h after transfection, COS-7 cells were stimulated with 3-isobutylmethyl-xanthine (1 mM)-containing medium, including the given reagent, for 30 min at 37 °C. In order to better detect cAMP reduction, 2 µM forskolin in IBMX-containing DMEM was added to cells for 5 min before washing and addition of collagen VI or pGPR126 in IBMX/DMEM medium. After stimulation, cells were lysed in LI buffer (PerkinElmer, Rodgau, Germany) and kept frozen at −20 °C until measurement. To measure cAMP concentration, the Alpha Screen cAMP assay kit (PerkinElmer) was used according to the manufacturer’s protocol. The accumulated cAMP was measured in 384-well white OptiPlate microplates (PerkinElmer) with the EnVision Multilabel Reader (PerkinElmer).

For direct quantitative measurement of inositol 1 phosphate (IP1), COS-7 cells were grown in DMEM as described. To perform the assay, Cisbio’s IP-One Tb kit (Cisbio, Codolet, France) was used. Then, 48 h after transfection, cells were stimulated for 60 min at 37 °C with 35 μL (96-well plates) or 150 µL (12-well plates) 1× IP1 stimulation buffer (Cisbio) containing the respective reagents. Cells were lysed in 30 μL (96-well plates) or 150 µL (12-well plates) lysis buffer (Cisbio) per well. The IP1 assay was terminated by adding lysis buffer to the cells. All probes were kept frozen at −20 °C until measurement. The accumulated IP1 was measured according to the manufacturer’s protocol in ProxiPlate-384 Plus microplates (PerkinElmer) with the EnVision Multilabel Reader (PerkinElmer).

For mechano-activation in both assays, the detached cell/medium mix was incubated for 30 min at 37 °C standing or shaking at 100 rpm (SH30/TH30; Edmund Bühler, Hechingen, Germany) with or without the addition of collagen VI (3 µg/mL).

#### 2.2.7. Data Analysis

Receptor activation was analyzed using paired *t*-test as indicated in each figure legend. *p* values < 0.05 were considered statistically significant (* *p* < 0.05; ** *p* < 0.01; *** *p* < 0.001). All statistical analyses were performed by GraphPad Prism version 6.00 for Windows (GraphPad Software, Inc. (San Diego, CA, USA)) or Microsoft Excel 2010 (Microsoft Corporation (Redmond, DC, USA)).

## 3. Results

### 3.1. Collagen VI Is a Novel Binding Partner of GPR126

ECM molecules such as collagen IV and laminin 211 have been shown to act as allosteric modulators of GPR126 [14,16,20]. To search for additional extracellular binding partners for this aGPCR, we conducted a pull-down experiment with the two NTF fragments of GPR126 as bait and extracts of sciatic nerves as ligand source. Mass spectrometry of the identified interacting molecules resulted in a list of ECM proteins (Appendix A). Based on our hypothesis that GPR126 in Schwann cells mediates cell–matrix interaction, we focused our attention on ECM proteins. Four ECM candidates were identified (Figure 2A). As a validation of our approach, a known ligand of GPR126 laminin 211 [16] was one of the four ECM protein candidates. Collagen VI is expressed by Schwann cells [21] and has been shown to play a role in the regulation of Schwann cell differentiation [22]. Recent studies have also implicated collagen VI in the maintenance of appropriate myelination in the peripheral nervous system (PNS), with Col6a1-knockout mice demonstrating hypermyelination of the sciatic nerve and decreased nerve conduction velocities [23]. Given the importance of collagen VI in the PNS, this seemed likely as a candidate for being a binding partner of GPR126.

To demonstrate the interaction between collagen VI with the NTF of GPR126 and to locate the binding site within the NTF, we performed a co-immunoprecipitation experiment with an N-terminal (NTF-31-348) and a C-terminal (NTF-446-807) portion of the NTF, both of them being C-terminally fused to an immunoglobulin Fc fragment (Figure 2B). As shown in Figure 2C, NTF-446-807 precipitated collagen VI, whereas NTF-31-348 and Fc showed no interaction. This binding location resembles the binding site of laminin 211 [16].

### 3.2. Collagen VI Is a Gi-Biased Ligand of GPR126

To investigate whether collagen VI is capable of inducing downstream signaling on GPR126, we incubated increasing concentrations of this protein with cells overexpressing the receptor. Collagen VI concentration–response curves revealed a decrease of cAMP upon increasing ligand concentrations without or with co-stimulation of 2.5 µM forskolin, suggesting Gi-protein coupling (Figure 3A). Addition of the Gi inhibitor pertussistoxin (PTX) could block this reduction in cAMP levels (Figure 3B,C) and using a Gqi chimeric protein that reroutes Gi-signaling to the Gq pathway [24] showed significant increase in IP1 levels upon the addition of collagen VI (Figure 3D). Of note, COS-7 cells endogenously express GPR126 [3], which results in partial activation also in empty vector control cells through collagen VI and the established peptide pGPR126 derived from the *Stachel* sequence. This activation, however, is abolished in a GPR126 T841 mutant that disables the encrypted agonist while retaining cell membrane expression [3] (Figure 3D), indicating that collagen VI activation is specific for GPR126 and requires the *Stachel* sequence.

### 3.3. Shaking Force Application Does Not Increase Gi-Signaling of GPR126

We previously showed that mechanical forces such as shaking and vibration can reverse Gs inhibition of laminin 211 to Gs activation that was more efficient than mechanical forces alone [16]. Thus, we tested the impact of shaking forces on collagen-VI-mediated activation of GPR126. While we observed the known significant increase in cAMP levels upon shaking of cells overexpressing GPR126, the combination of collagen VI and shaking did not significantly reduce them (Figure 4A). In line with that, Gqi-mediated increase in IP1 levels was significantly increased over basal GPR126 IP1 levels through force or collagen VI addition alone or in combination but no additional activation was seen (Figure 4B).

## 4. Discussion

GPR126 is a Gs- and Gi-protein-coupled receptor that guides Schwann cell development and potentially other tissue functions through fine-tuning of cAMP levels in the cell. Activation of the receptor can be achieved through different stimuli, including interactions with extracellular ligands, such as collagen IV and laminin 211, or mechanical forces. There are multiple reports on the effect of these interactions [14,16], which all indicated a specific modulation of the Gs protein-mediated signaling. The possibility for GPR126 to actively engage Gi proteins has so far only been shown for modulators that can directly interact with the 7 TMD, such as the *Stachel*-derived peptides [3] or the small molecule compounds progesterone or 17-hydroxyprogesterone [13]. The effect of mechanical stimuli such as shaking on Gi activation have not been studied for GPR126. In our study, we identify collagen VI as a new ligand that binds the N terminus of GPR126 at a similar location as the previously identified interaction partner, laminin 211 [16] (Figure 2). Both ligands bind to a truncated portion of the N terminus, which roughly corresponds to the GAIN domain. As the truncation is designed to start at the endogenous furin site within the N terminus, we believe that this will not preclude correct folding of the protein. Both laminin 211 and collagen VI reduce GPR126-mediated cAMP levels; the mechanism behind these, however, differs decisively. Laminin 211 inhibits Gs, which can be reversed through the addition of mechanical forces. Collagen VI activates Gi, which cannot be altered by shaking of cells (Figure 3 and Figure 4). In most activation scenarios of GPR126, cAMP increase is the predominant observation. This also holds true for the constitutively active CTF mutants or *Stachel*-derived peptides, which represent the full engagement of the encrypted agonist sequence within its binding pocket. Ligand-mediated activation has been explained with a scenario in which binding partner engagement or mechanical force application release the NTF, thereby leaving the constitutively active CTF behind to induce maximum signaling capacity. As collagen VI specifically activates Gi protein, resulting in a reduction of cAMP levels, the question arises of how this can be mediated on a structural level. As the structural basis for G protein bias of GPR126 remains to be solved, the mechanism behind the collagen-VI-mediated Gi bias can only be speculated about. It would be conceivable that either the *Stachel* is prebound in the binding pocket and can therefore be modified in its position or there is a partial release from the GAIN domain that allows for distinct interactions with the binding pocket. Future studies will have to tackle this question.

On a functional level, the GPR126–collagen VI interaction could play a role in Schwann cell myelination similar to laminin 211 and collagen IV as collagen VI is another component of the basal lamina. Col6a1-knockout mice show hypermyelination with reduced nerve conduction velocity and motor coordination [23]. It can be speculated that this was caused by a loss of cAMP suppression; however, cAMP levels were not measured in this study and it is still unknown how excessive cAMP influences Schwann cell biology. Apart from this, an overlapping expression pattern of GPR126 and collagen VI in triple negative breast cancer can be found, and a recent study indicated that GPR126-mediated Gi signals in these cells propagate cancer progression [13,25].

## 5. Conclusions

We identified collagen VI as a new Gi-biased ligand of GPR126, which represents the first N-terminal ligand of an aGPCR that can mediate a G-protein preference. While the structural basis for this functional interaction remains to be solved in the future, it opens new insights into the different activation scenarios that can occur within this GPCR class and will guide further our understanding of the physiological role of GPR126 and its contribution to diseases.

## Figures and Tables

**Figure 1 cells-12-01551-f001:**
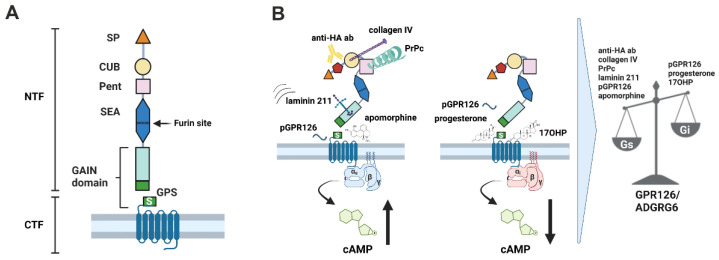
Overview of the GPR126/ADGRG6 and ways to activate Gs and Gi signaling pathways through it. (**A**) The N terminus of GPR126 contains the CUB (pink eclipse), a pentraxin domain (Pent, purple square) and the Sperm protein, Enterokinase and Agrin (SEA) domain (blue polygon), including the furin site (dotted line). The highly conserved GPCR autoproteolysis-inducing (GAIN) domain (turquois rectangle) contains the GPCR proteolysis site (GPS) at which the receptor is cleaved into an N-terminal fragment (NTF) and a C-terminal fragment (CTF). The CTF harbors the *Stachel* sequence (S, green rectangle) at its extracellular portion. (**B**) GPR126 can signal through Gs protein by stimulation with either *Stachel*-sequence-derived peptides (pGPR126), small molecule apomorphine, anti-HA antibody (anti-HA ab) targeting an N-terminal inserted HA tag or through its ligands collagen IV, prion protein (PrPc) or laminin 211 in combination with mechanical forces. The small molecules progesterone and 17OHP, but also pGPR126, activate GPR126 via the Gi signaling pathway. Figure was created with biorender.

**Figure 2 cells-12-01551-f002:**
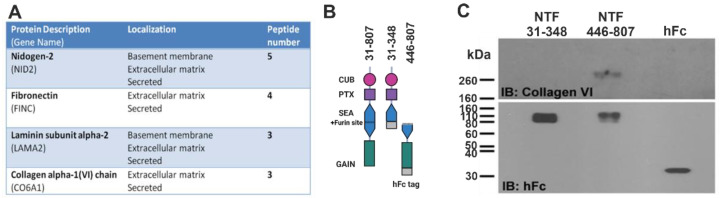
Collagen VI binds to the NTF of GPR126. (**A**) Summary of mass spectrometry results. Total number of peptide sequences specific for binding to GPR126-NTF. (**B**) NTF protein parts used for co-IP. (**C**) co-IP of GPR126 NTF with wild-type sciatic nerves, probed with anti-collagen VI antibody, demonstrates binding of collagen VI with GPR126 NTF-446-807 fragment. Detected bands for the fusion proteins are slightly bigger than theoretically estimated, indicating potential glycosylation. IB, immunoblotting.

**Figure 3 cells-12-01551-f003:**
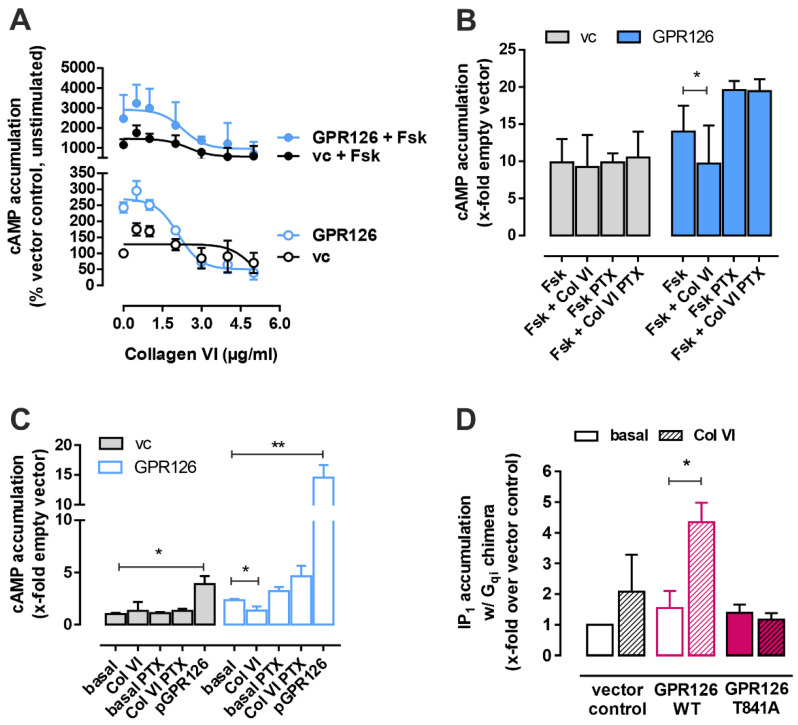
Collagen VI is a new ligand of GPR126 with activating properties in the Gi-signaling pathway. (**A**) Concentration–response curves upon stimulation of wild-type GPR126 or vector control with collagen VI are shown as % of unstimulated vector control cAMP level (vc: cAMP level: 1.46 ± 0.3 nM = 100%). cAMP accumulation is displayed in the presence or absence of 2.5 µM forskolin co-stimulation in COS-7 cells as means ± SEM of three independent experiments each performed in triplicates. (**B**,**C**) Pretreatment with PTX abolishes reduction in cAMP through incubation with collagen VI (2 µg/mL) in cells stimulated (**B**) with or (**C**) without Fsk (2 µM). Of note, stimulation with the agonistic peptide pGPR126 (1 mM) shows significant increase of cAMP in assay conditions without Fsk co-stimulation (**C**). Data are shown as fold over unstimulated vector control (vc: cAMP level: 4.76 ± 1.45 nM) as means ± SEM of four independent experiments each performed in triplicates. Statistics were performed using paired *t*-test; * *p* < 0.05, ** *p* < 0.01. (**D**) COS-7 cells were transfected with wild-type, T841A *Stachel* mutant GPR126 or vector control. Accumulation of IP1 with Gqi chimera was measured after treatment of the transfected cells with or without collagen VI (2 µg/mL). Data are shown as fold over unstimulated vector control (vc: IP1 level: 6.37 ± 3.5 nM) as means ± SEM of three independent experiments each performed in triplicates. Statistics were performed using paired *t*-test; * *p* < 0.05.

**Figure 4 cells-12-01551-f004:**
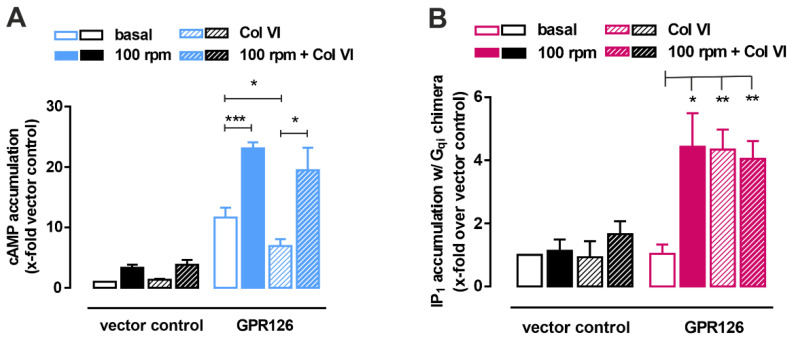
Collagen-VI-mediated Gi activation is not amplified through additional shaking forces. (**A**) Accumulation of cAMP or (**B**) IP1 with Gqi chimera was measured after treatment of the transfected cells without mechanical forces (basal) or detachment of the transfected cells followed by mechano-activation at 100 rpm or through the combined application of mechanical forces and collagen VI (2 µg/mL Col VI). Vector control (vc) served as negative control (vc; cAMP level: 3.26 ± 0.2 nM; vc; IP1 level Gqi: 8.48 ± 3.74 nM). Data are given as x-fold over vector control as means ± SEM of at least three independent experiments each performed in triplicates. Statistics were performed using paired *t*-test; * *p* < 0.05, ** *p* < 0.01, *** *p* < 0.001.

## Data Availability

The data generated and analyzed for the current study are available from the corresponding author (I.L.).

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
