# Peer review of "Collagen VI Is a Gi-Biased Ligand of the Adhesion GPCR GPR126/ADGRG6"

_cells, 2023, doi:10.3390/cells12111551_

Round 1

Reviewer 1 Report

      This is a seemingly straightforward report that identifies collagen VI as ligand of the adhesion receptor GPR126 that induces Gi-mediated signaling, in contrast to other known ligands of this receptor that promote Gs-mediated signaling. This finding could have implications for understanding and controlling the diverse apparent signaling functions of GPR126. The overall conclusions appear to be reasonable and potentially worth publishing, but I have the following concerns about interpretation of the experiments, the lack of experimental detail, lack of certain controls, and the clarity of the manuscript (not listed in order of importance).

1) the description of the initial pulldowns and mass spectroscopy is inadequate:

i) The conditions for the expression, purification, and biotinylation of the  GPR126NTF-mFc-biotag are not described.

ii) The conditions for the pull down, for example concentrations of purified protein and lysate and conditions for streptavidin binding, are not described.

iii) The gel from which bands were cut for mass spec analysis should be shown. There is no information on what bands were chosen for excision or why.

iv) No procedures for mass spec or subsequent analysis are described. What types of pre-separations, instruments and analyses were used? How were peptides selected for sequence analysis? What statistical metrics were used for peptide identification?

v) How were proteins chosen for display in Fig. 2a? Were these all the proteins that could be identified (very unlikely using modern instrumentation)? Were any other proteins tested for effects on GPR126?

vi) Why is the number of peptides identified from the large proteins in Fig. 2a so small? If Collagen VI is interacting with GPR126, why were multiple chains of collagen VI not detected?

vii) These concerns, in part, reflect common concerns about potential cherry-picking of mass-spec identified interactions that suit the expectations of experimenters.

2) Fig. 1 needs clarification:

i) The turquoise rectangle is hard to find- and what are the blips inside it?

ii) What is the meaning of the upward arrow associated with laminin 211?

iii) The legend states that the green rectangle Stachel sequence follows the GPS domain, but the original covalent linkage is not shown.

iv) What is the GPS?

v) The GAIN domain should be labeled.

vi) It is hard to distinguish proteins that bind to GPR126 from the different NTF domains.

vii) What is meant by showing the Gi/Gs as a balance? Do they necessarily oppose each other? Does the lower pan for Gs mean that Gs signaling is stronger? Is this supported by evidence? 

3) The left hand construct in Fig. 2B should be labeled as the full-length NTF. Also, it is confusing that the Fc tag is not to scale, since it is almost as big as the NTF fragment.  

4) Regarding Fig. 2c, there should be some indication that only three lanes were loaded, if that is, in fact the case.

5) A co-IP like that shown in Fig. 2c should also show loading controls and input lanes so that it is apparent how much of the input protein was precipitated. It should also contain an indication of the mode of purification and concentration of the purified protein used for the pull-down.

 6) There appears to be something missing from the description of the pulldown and co-IP experiments in Fig. 2C. The protocol sounds like the lysates were incubated with purified proteins, then immediately denatured and separate by SDS PAGE. How is this a pull-down?

 7) In Fig. 3a, why does treatment of vector control (No GPR126) with collagen VI cause what appears to be a significant reduction in cAMP? This raises the question of whether the action of collagen VI might be through some mechanism other than GPR126 that is further enhanced by expression of GPR126.

8) Fig 3B does not indicate the concentration of collagen VI used.

 9) Fig. 3B should include control samples in cells expressing GPR126 that were not treated with PTX under identical conditions. This is a serious omission in view of the claim that PTX decreases Gi signaling.

 10) In Fig. 3C, why does collagen VI induce almost as large a percent increase in IP1 in the vector control as in the GPR126 samples? The lack of a “significant” increase is not meaningful with such a large error of measurement.

11) The description of Fig. 4A is confusing. The decrease in cAMP caused by collagen VI (concentration not specified) in the absence of shaking seems insignificant, which is surprising in view of the claimed activation of Gi and the data in Fig. 3A. On the other hand, the decrease in cAMP in cells collagen VI-treated cells with shaking does seem significant (despite the lack of an asterisk), apparently contradicting the claim that “While we observed the known significant increase in cAMP levels upon shaking of cells overexpressing GPR126, the combination of collagen VI and shaking did not significantly reduce them.” Again, the concentration of collagen VI is not given.

 12) The statement “Gqi-mediated increase in IP1 levels was similarly significantly increased over basal GPR126 IP1 levels independent of applied force or collagen VI addition (Fig. 4B),” appears to be incorrect or confusing. Either shaking or collagen VI, individually or separately, drastically increases IP1 levels.

 13) The lack of co-IP between collagen VI and NTF-31-348 is not definitive evidence for the involvement of the C-terminal portion of the NTF in the interaction, since truncation may prevent proper folding and stability of the remaining NTF domains.

14) I could not find any description of the vector expressing N- and C-terminally tagged GPR126 for functional assays. Do the tags affect receptor expression or function? Along similar lines, GPR126 has multiple splice variants- which form was used for these experiments?

15) Could overexpression of receptor and/or Gqi protein constructs or the use of COS cells drive non-physiological functional outputs?  Presumably the Gs responses were from endogenous G protein. Is the Gqi construct a reliable way of detecting authentic physiologically relevant Gi responses?  

Author Response

We want to thank the reviewer for taking his valuable time to review our manuscript. A response for each point is outlined below.

This is a seemingly straightforward report that identifies collagen VI as ligand of the adhesion receptor GPR126 that induces Gi-mediated signaling, in contrast to other known ligands of this receptor that promote Gs-mediated signaling. This finding could have implications for understanding and controlling the diverse apparent signaling functions of GPR126. The overall conclusions appear to be reasonable and potentially worth publishing, but I have the following concerns about interpretation of the experiments, the lack of experimental detail, lack of certain controls, and the clarity of the manuscript (not listed in order of importance).

1) the description of the initial pulldowns and mass spectroscopy is inadequate:

Thank you very much for pointing this decisive omission out. We added all missing parts to the Materials and Methods Section. We apologize for the initial inaccurate description of MS. We used the whole lane for mass spec analysis. We now provide the entire data as supplemental table S1. We applied two filters: (1) only extracellular matrix proteins were included based on our hypothesis that GPR126 in Schwann cells mediates cell-matrix interaction; and (2) those proteins with equal to or more than 3 unique peptides detected were listed in the table in Fig. 2.

  1. i) The conditions for the expression, purification, and biotinylation of the GPR126NTF-mFc-biotag are not described.
  2. ii) The conditions for the pull down, for example concentrations of purified protein and lysate and conditions for streptavidin binding, are not described.

iii) The gel from which bands were cut for mass spec analysis should be shown. There is no information on what bands were chosen for excision or why. 

  1. iv) No procedures for mass spec or subsequent analysis are described. What types of pre-separations, instruments and analyses were used? How were peptides selected for sequence analysis? What statistical metrics were used for peptide identification?
  2. v) How were proteins chosen for display in Fig. 2a? Were these all the proteins that could be identified (very unlikely using modern instrumentation)? Were any other proteins tested for effects on GPR126?
  3. vi) Why is the number of peptides identified from the large proteins in Fig. 2a so small? If Collagen VI is interacting with GPR126, why were multiple chains of collagen VI not detected?

vii) These concerns, in part, reflect common concerns about potential cherry-picking of mass-spec identified interactions that suit the expectations of experimenters.

2) Fig. 1 needs clarification:

  1. i) The turquoise rectangle is hard to find- and what are the blips inside it? Thank you very much for pointing out this omission. We added a figure with the respective domain labeling as figure 1A.
  2. ii) What is the meaning of the upward arrow associated with laminin 211? The arrow was indicative of the pulling forces required for laminin-211 induced activation. This detail is, however, not necessary and we deleted it from the figure.

iii) The legend states that the green rectangle Stachel sequence follows the GPS domain, but the original covalent linkage is not shown.

As the autoproteolysis of aGPCRs already occurs within the endoplasmatic reticulum (PMID: 15150276, PMID: 34022221), there will be no covalent bond at the cell membrane as we show the receptor here.

  1. iv) What is the GPS? Thank you very much for pointing out this omission. We added a figure with the respective domain labeling as figure 1A, including the GPS and a little explanation of it in the figure legend.
  2. v) The GAIN domain should be labeled. Thank you very much for pointing out this omission. We added a figure with the respective domain labeling as figure 1A.
  3. vi) It is hard to distinguish proteins that bind to GPR126 from the different NTF domains. Thank you for pointing this out. We changed the color style to allow for better discrimination.

vii) What is meant by showing the Gi/Gs as a balance? Do they necessarily oppose each other? Does the lower pan for Gs mean that Gs signaling is stronger? Is this supported by evidence? 

As Gs and Gi proteins both modulate adenylylcyclase activation in an opposing manner they are naturally balancing each other. Of course, both proteins can have different roles in signal transduction but these have not been shown to our knowledge. As the scale represents a summary of the shown cAMP modulation it should be clear that we refer to this signaling effect. As it is a scale, the lower pan means stronger impact. All listed modulators are published and we cite all studies in the text see p. 2 ll. 47-56.

3) The left hand construct in Fig. 2B should be labeled as the full-length NTF. Also, it is confusing that the Fc tag is not to scale, since it is almost as big as the NTF fragment.

Thank you for pointing this out. As this construct does not include the signal peptide, it should be labeled as “31-807”. The figure has symbol character to illustrate the domain structure and does not represent actual sizes.

4) Regarding Fig. 2c, there should be some indication that only three lanes were loaded, if that is, in fact the case.

As shown in the attached original gel, only three lanes were loaded and were separated with one open lane in between.

5) A co-IP like that shown in Fig. 2c should also show loading controls and input lanes so that it is apparent how much of the input protein was precipitated. It should also contain an indication of the mode of purification and concentration of the purified protein used for the pull-down.

We normally provide loading control for Western blot from cell lysates or tissue homogenates. However, for Western blot of co-IP immunocomplex, we load the whole immunoprecipitated protein complex. We provide all original whole membrane blots in the respective submission section.

6) There appears to be something missing from the description of the pulldown and co-IP experiments in Fig. 2C. The protocol sounds like the lysates were incubated with purified proteins, then immediately denatured and separate by SDS PAGE. How is this a pull-down?

We added all missing parts to the Materials and Methods Section.

7) In Fig. 3a, why does treatment of vector control (No GPR126) with collagen VI cause what appears to be a significant reduction in cAMP? This raises the question of whether the action of collagen VI might be through some mechanism other than GPR126 that is further enhanced by expression of GPR126.

COS-7 cells endogenously express GPR126 (PMID: 25533341) but in small amounts that are only functionally relevant when large concentrations of stimuli are employed, which is why it is not feasible to only use the endogenous receptor. We added this information with the respective reference to the paper p. 5/6 ll. 247-250. That’s why we were using a medium concentration of 2 mg/ml of collagen VI for all further specification experiments, which does not give a significant signal for empty vector control while showing significant effects in GPR126 transfected cells. Specificity of the collagen VI-mediated Gi signaling is further shown through the expression of a GPR126 mutant with a disrupted tethered agonist sequence (see Fig. 3D).

8) Fig 3B does not indicate the concentration of collagen VI used. Thank you very much for pointing out this omission.

We added this information to the figure legend.

 9) Fig. 3B should include control samples in cells expressing GPR126 that were not treated with PTX under identical conditions. This is a serious omission in view of the claim that PTX decreases Gi signaling.

With this figure being right next to the concentration response curve showing cAMP reduction we thought this would be self-explanatory but we of course have all the controls available and are happy to show the significant reduction in the bar graphs (Fig. 3B/C for effects with and without Fsk, respectively).

 10) In Fig. 3C, why does collagen VI induce almost as large a percent increase in IP1 in the vector control as in the GPR126 samples? The lack of a “significant” increase is not meaningful with such a large error of measurement.

COS-7 cells endogenously express GPR126 (PMID: 25533341) but in small amounts that are only functionally relevant when large concentrations of stimuli are employed, which is why it is not feasible to only use the endogenous receptor. We added this information with the respective reference to the paper p. 6/7 ll. 247-250. That’s why we were using a medium concentration of 2 mg/ml of collagen VI for all further specification experiments, which does not give a significant signal for empty vector control while showing significant effects in GPR126 transfected cells. Specificity of the collagen VI-mediated Gi signaling is further shown through the expression of a GPR126 mutant with a disrupted tethered agonist sequence (see Fig. 3D).

11) The description of Fig. 4A is confusing. The decrease in cAMP caused by collagen VI (concentration not specified) in the absence of shaking seems insignificant, which is surprising in view of the claimed activation of Gi and the data in Fig. 3A.

The concentration of collagen VI is given in the figure legend, which states the same conditions for A and B. The decrease in cAMP through collagen VI is in the same range as shown in Figure 2A with approx. 20% reduction. The direct comparison with highly activating conditions can make the reduction appear insignificant as is indicated by using an unbiased ANOVA test for statistical analysis. To allow for better comparison we applied now a paired T-test, which shows significant differences in shaking activation with and without collagen VI and the significant reduction through collagen VI on static cell cultures.

On the other hand, the decrease in cAMP in cells collagen VI-treated cells with shaking does seem significant (despite the lack of an asterisk), apparently contradicting the claim that “While we observed the known significant increase in cAMP levels upon shaking of cells overexpressing GPR126, the combination of collagen VI and shaking did not significantly reduce them.” Again, the concentration of collagen VI is not given.

Even the now applied paired T-test does not show a significant reduction in collagen VI/ shaking combination. We repeated these experiments to see if this would change but it only activated a little more than before. The overall statement remains the same: the combination of collagen VI and shaking is not resulting in significantly reduced cAMP levels compared to just shaking. The concentration of collagen VI is given in the figure legend, which states the same conditions for A and B.

12) The statement “Gqi-mediated increase in IP1 levels was similarly significantly increased over basal GPR126 IP1 levels independent of applied force or collagen VI addition (Fig. 4B),” appears to be incorrect or confusing. Either shaking or collagen VI, individually or separately, drastically increases IP1 levels.

We changed the sentence accordingly see p. 6 ll.274/275.

13) The lack of co-IP between collagen VI and NTF-31-348 is not definitive evidence for the involvement of the C-terminal portion of the NTF in the interaction, since truncation may prevent proper folding and stability of the remaining NTF domains.

There is of course always a chance that truncation may alter the folding of the remaining protein. However, the constructs used here are divided at a natural cleavage site, which is the furin site within the SEA domain (see Figs. 1A/2B). Thus, the N terminus of GPR126 is pretty much always split at exactly this position (see PMID: 15189448). If one would want to see the whole NTF this furin site would need to be mutated, rendering the construct truly artificial. Further, the GAIN domain has been shown multiple times to be able to fold in the almost exact same conformation on its own or in combination with additional N-terminal domains. This is possible because the N-terminus of aGPCRs is composed of several domains, which can fold independently. The N terminus of GPR126 has been shown in a crystal structure (PMID: 31924782), where the GAIN domain looks just like the isolated GAIN domains of Lat-1 and Bai3 (PMID: 22333914).

14) I could not find any description of the vector expressing N- and C-terminally tagged GPR126 for functional assays. Do the tags affect receptor expression or function? Along similar lines, GPR126 has multiple splice variants- which form was used for these experiments?

We used the same construct that we already published and characterized several times. Thank you for pointing out that we missed to reference that. We added this now to the methods section p. 4.

15) Could overexpression of receptor and/or Gqi protein constructs or the use of COS cells drive non-physiological functional outputs?  Presumably the Gs responses were from endogenous G protein. Is the Gqi construct a reliable way of detecting authentic physiologically relevant Gi responses?

The use of chimeric G proteins has been debated with respect to potential unphysiological outputs. That’s why we provide a combined assay setup, where we show cAMP reduction and its reversibility through the addition of PTX (a specific inhibitor of endogenous Gi proteins) in addition to the active recruitment of the Gqi chimera.

Reviewer 2 Report

In their manuscript, authors identify collagen VI as a novel interacting protein for the N-terminal domain of GPR126 by a proteomics screening approach and subsequent biochemical hit validation. They further show that collagen VI binds to a region of GPR126 that is similar to that of the laminin 211 bindin site. Moreover, they demonstrate that collagen VI induces predominantly Gi-mediated signaling of GPR126.

The study is well conducted and the data is presented clearly and concisely.

Concerns:

Collagen VI seems to reduce cAMP formation with similar efficacy also in the absence of GPR126 (the fit of the cAMP curve of vector-control in absence of Fsk also seems to follow a similar slope when the first data point is left out). Can authors exclude that collagen VI is impacting cAMP formation through a mechanism independent of GPR126?

To state that collagen VI is a truly biased ligand for GPR126, authors should provide cAMP measurements with a reference ligand (e.g. laminin 211) under the same conditions.

In addition to collagen VI, authors identify at least three additional proteins with higher abundance in mass spectrometry. Were these the only proteins found by mass spectrometry?

The methods section has some inconsistencies and lacks several details:

 - Purification of GPR126NTF fragments used for pull down and Co-IP and Co-IP experiments are detailed very sparsly. Which column purification was used? How was the precipitation conducted? Also, more details of mass spectrometry should be provided.

- L91: How is the GPR126NTF-mFc-Biotag construct designed? Did mFc control proteins also contain a biotag?

- L92: Please specify species and strain from which sciatic nerves were prepared?

- L 126: Heading is called „Pull-down and Co-Immunoprecipitation“ but only contains description of Co-IP.

- L110: I assume, numbers following the GPR126-NTF fragments correspond to residues. Please write out.

- L123 ff: The Gqi4 chimera needs to be referenced.

Fig. 1 is a bit confusing as inclusion of the HA-tag/ anti-HA antibody suggests it to be a natural mechanism for GPR126 activation.

Original image to Fig. 2C is identical to the processed figure.

Author Response

We want to thank the reviewer for taking his valuable time to review our manuscript. A response for each point is outlined below.

In their manuscript, authors identify collagen VI as a novel interacting protein for the N-terminal domain of GPR126 by a proteomics screening approach and subsequent biochemical hit validation. They further show that collagen VI binds to a region of GPR126 that is similar to that of the laminin 211 bindin site. Moreover, they demonstrate that collagen VI induces predominantly Gi-mediated signaling of GPR126.

The study is well conducted and the data is presented clearly and concisely.

Thank you very much for your positive evaluation of our study.

Concerns:

Collagen VI seems to reduce cAMP formation with similar efficacy also in the absence of GPR126 (the fit of the cAMP curve of vector-control in absence of Fsk also seems to follow a similar slope when the first data point is left out). Can authors exclude that collagen VI is impacting cAMP formation through a mechanism independent of GPR126?

COS-7 cells endogenously express GPR126 (PMID: 25533341) but in small amounts that are only functionally relevant when large concentrations of stimuli are employed, which is why it is not feasible to only use the endogenous receptor. We added this information with the respective reference to the paper p. 6/7 ll. 247-250. That’s why we were using a medium concentration of 2 mg/ml of collagen VI for all further specification experiments, which does not give a significant signal for empty vector control while showing significant effects in GPR126 transfected cells. Specificity of the collagen VI-mediated Gi signaling is further shown through the expression of a GPR126 mutant with a disrupted tethered agonist sequence (see Fig. 3D).

To state that collagen VI is a truly biased ligand for GPR126, authors should provide cAMP measurements with a reference ligand (e.g. laminin 211) under the same conditions.

We agree that this would be a very interesting experiment. However, due to the short revision time it is not feasible for us to perform these experiments as shipping times for collagens (that are not collagen I) are very long. A statement towards collagen VI being a biased agonist is, however, still accurate as the bias refers to the specific downstream signaling that is induced. Multiple studies by us and others have shown that Gs and Gi- coupling occurs for GPR126 (all cited in the introduction). Collagen VI shows Gi-coupling preference over Gs-coupling as seen in cAMP inhibition, that is reversible through the Gi-inhibitor PTX, and in IP1 accumulation employing a Gqi chimera. This makes collagen VI a Gi-biased ligand.

In addition to collagen VI, authors identify at least three additional proteins with higher abundance in mass spectrometry. Were these the only proteins found by mass spectrometry?

As in most cases, our mass spectrometry analysis revealed many protein candidates (supplemental table S1). The key in our experience is to apply appropriate filters in mining the data. In this case, we focused extracellular protein  based on our hypothesis that Schwann cell GPR126 mediates cell-matrix interaction. Additionally, we only listed proteins with equal or more than 3 unique peptides identified in mass spectrometry.

The methods section has some inconsistencies and lacks several details: 

Thank you very much for pointing this decisive omission out. We added all missing parts to the Materials and Methods Section.

 - Purification of GPR126NTF fragments used for pull down and Co-IP and Co-IP experiments are detailed very sparsly. Which column purification was used? How was the precipitation conducted? Also, more details of mass spectrometry should be provided.

- L91: How is the GPR126NTF-mFc-Biotag construct designed? Did mFc control proteins also contain a biotag?

- L92: Please specify species and strain from which sciatic nerves were prepared?

- L 126: Heading is called „Pull-down and Co-Immunoprecipitation“ but only contains description of Co-IP.

- L110: I assume, numbers following the GPR126-NTF fragments correspond to residues. Please write out.

Thank you for the suggestion. We have since spelled out as “NTF-31-348 and NTF-446-807” in the text.

- L123 ff: The Gqi4 chimera needs to be referenced.

Thank you for pointing this out. We added the reference.

Fig. 1 is a bit confusing as inclusion of the HA-tag/ anti-HA antibody suggests it to be a natural mechanism for GPR126 activation.

We changed the figure legend to: “Overview of the human GPR126/ADGRG6 and ways to activate Gs and Gi signaling pathway through it.” This should clarify that not only natural mechanisms are shown.

Original image to Fig. 2C is identical to the processed figure.

Thank you for pointing this out. We made a mistake before and added now all necessary original blots.

Round 2

Reviewer 1 Report

    While the manuscript is improved, I do not feel that all of the issues raised in the initial review have been addressed and some of the revisions raise new questions. Evaluation of the revised manuscript was impaired by the lack of a fully tracked version of the revised document.

 1) The description of the biotinylation reaction states that 1ml of enzyme was used but provides no indication of the enzyme concentration or activity, nor does it state the total volume of the reaction.

 2) The description of the mass spectrometry gives no indication of instrument used, the acquisition mode, or the false discovery rate.

 3) Regarding critical parameters describing the pull-down experiments, the manuscript provides no indications of amounts of beads, incubation volumes, washing and blocking conditions, etc.

 4) The inclusion of Table S1 is an improvement.

 5) The authors have not addressed the question from the previous review of why only 3 peptides could be identified from the ~1700 amino acids of collagen IV a1-chain and why there was no detection of accompanying chains of collagen IV.

 6) Figs. 1 and 2 are improved.

 7) I presume that the legend for Fig. 2C should read ”probed with anti-collagen VI”.

 8) I do not understand the authors’ response regarding the lack of loading controls in Fig. 2C. The fact that they loaded the whole sample from IP does not address the question. I do not know what the “respective submission section” refers to. These controls should be shown in the published paper.

 9) The description of experimental conditions for the co-IPs, while improved, is still inadequate. How much of the purified protein was present, how much beads (and from where), what washes were performed and under what conditions? Were samples heated in SDS, etc?  

 10) The authors have provided useful clarification on the role of endogenous GPR126 in COS cells. Of course, this would be best addressed by experiments with cells that are knocked out for GPR126.  The experiments with the T841A mutant show a relevant dominant negative effect but do not actually prove that the endogenous effects are due to GPR126.

 11) If GPR126 is Gi coupled, it is not clear why expression of the protein in the absence of Col IV increases cAMP production in Figs. 3A-C and 4A. It is also not clear why addition of ligand in the presence of PTX in Fig.  3C increases cAMP production. These results seem to suggest some Gs-coupled ligand-activated and constitutive activity.

 12) In Fig. 3C, in the absence of Col IV, PTX strongly increases basal cAMP production in GPR126-epxressing cells but not vector controls. PTX also significantly increases cAMP production in GPR126-expressing cells in Fig. 3B in the presence of forskolin in the presence and absence of Col IV. These results suggests that there is constitutive Gi-mediated GPR126 signaling in these cells.

 12) Several aspects of Fig. 3 are still unclear. I presume the vertical axis of both 3A and 3B refers to vector alone in the absence of forskolin. But, there also appears to be some discrepancy between the two panels Fig. 3A and 3B. In presence of forskolin and the absence of col IV, in panel A, GPR126 is more than 2 X the forskolin treated vector. In Panel B, in the absence of col IV, the cAMP is less than 1.5 X the vector control. This discrepancy is on the same order of magnitude as the col IV effect.

13) The data in Fig. 4 has changed from the initial submission. I assume that this is based on repeating the experiments.

14) The paired t-test is only applicable when the samples are actually experimental pairs. Was this the case?

 15) Something seems to be wrong with the application of the single asterisk to the comparison of the right-hand two bars in Fig. 4. The difference between these two bars seems more significant than any of the other comparisons.

 16) I do not understand this sentence: “We could previously show that mechanical forces such as shaking and vibration can reverse Gs-inhibition of laminin-211 to Gs-activation that was more efficient than mechanical forces alone [16].”

  17) The manuscript should include a discussion by the authors of their stated reasons from believing that truncation of the protein will not preclude correct folding.

Author Response

We want to thank the reviewer for taking his valuable time to review our manuscript. A response for each point is outlined below.

 1) The description of the biotinylation reaction states that 1ml of enzyme was used but provides no indication of the enzyme concentration or activity, nor does it state the total volume of the reaction.

Please see the full step-by-step protocol as supplemental material.

 2) The description of the mass spectrometry gives no indication of instrument used, the acquisition mode, or the false discovery rate.

The LTQ Orbitrap Velos Pro ion-trap mass spectrometer (Thermo Fisher Scientific, Waltham, MA) was used. Peptides were detected, isolated, and fragmented to produce a tandem mass spectrum of specific fragment ions for each peptide. Peptide sequences (and hence protein identity) were determined by matching protein databases with the acquired fragmentation pattern by the software program, Sequest (Thermo Fisher Scientific, Waltham, MA).  All databases include a reversed version of all the sequences and the data was filtered to between a one and two percent peptide false discovery rate.  A total of four independent experiments were performed. The false discovery rates are 0.48%, 0.12%, 0.65% and 0.28%.

 3) Regarding critical parameters describing the pull-down experiments, the manuscript provides no indications of amounts of beads, incubation volumes, washing and blocking conditions, etc.

Please see the full step-by-step protocol as supplemental materiel.

 4) The inclusion of Table S1 is an improvement.

Thank you.

 5) The authors have not addressed the question from the previous review of why only 3 peptides could be identified from the ~1700 amino acids of collagen IV a1-chain and why there was no detection of accompanying chains of collagen IV.

Collagen VI is an extracellular matrix (ECM) protein. It is challenging to detect any ECM protein because they often cross-link with other ECM proteins. We are delighted to be able to detect 3 peptides of collagen VI.

 6) Figs. 1 and 2 are improved.

Thank you.

 7) I presume that the legend for Fig. 2C should read ”probed with anti-collagen VI”.

That is correct. Thank you.

 8) I do not understand the authors’ response regarding the lack of loading controls in Fig. 2C. The fact that they loaded the whole sample from IP does not address the question. I do not know what the “respective submission section” refers to. These controls should be shown in the published paper.

A loading control in a Co-IP assay is technically not possible as the Co-IP only allows for the detection of specific interaction partners. What would be the neutral control for this? We state the amount of protein used in this assay and we are not making any statement regarding a quantitative difference.

 9) The description of experimental conditions for the co-IPs, while improved, is still inadequate. How much of the purified protein was present, how much beads (and from where), what washes were performed and under what conditions? Were samples heated in SDS, etc?  

Please see the full step-by-step protocol as supplemental materiel.

 10) The authors have provided useful clarification on the role of endogenous GPR126 in COS cells. Of course, this would be best addressed by experiments with cells that are knocked out for GPR126.  The experiments with the T841A mutant show a relevant dominant negative effect but do not actually prove that the endogenous effects are due to GPR126.

The dominant negative effect of the mutant GPR126 provides a good indication for receptor specific effects of collagen VI. Knock-out of the receptor either with siRNA or genetically using Crispr Cas9 can also induce dominant negative effects that could mask endogenous responses. As such all these attempts would always just provide a hint. On the other hand, collagen VI can bind to many surface antigens and why would it not activate another molecule? We show with the mutant that collagen VI reduces cAMP levels only in a functionally expressed GPR126, which would normally preferentially increase cAMP levels through Gs. And that is the story of the paper.

 11) If GPR126 is Gi coupled, it is not clear why expression of the protein in the absence of Col IV increases cAMP production in Figs. 3A-C and 4A. It is also not clear why addition of ligand in the presence of PTX in Fig.  3C increases cAMP production. These results seem to suggest some Gs-coupled ligand-activated and constitutive activity. 12) In Fig. 3C, in the absence of Col IV, PTX strongly increases basal cAMP production in GPR126-epxressing cells but not vector controls. PTX also significantly increases cAMP production in GPR126-expressing cells in Fig. 3B in the presence of forskolin in the presence and absence of Col IV. These results suggests that there is constitutive Gi-mediated GPR126 signaling in these cells.

As we explain and reference in the introduction of the paper, GPR126 can couple to both, Gs and Gi in a constitutively manner. The Gs pathway, however, is usually predominant. So far, only progesterone was shown to switch this Gs preference to a Gi preference. While progesteron binds in the 7TMD binding pocket, collagen VI mediates the Gi preference through interaction with the N terminus through the tethered agonist. To stress once more that Gs preference is also seen in basal activation levels, we added the word “basal” to the respective sentence in the introduction (p. 1, l. 40), which now reads as follows:: “This basal, peptide or mutation-induced activation usually results in an increase in cAMP levels, which indicates Gs-coupling [3].”

This basic capacity of GPR126 to couple Gs and Gi results in the observed increase in cAMP upon PTX addition. The additional not significant but observable increase in cAMP levels upon addition of col VI and PTX could indicate that in the absence of a functional Gi protein also Gs could couple again in these conditions but this can only be speculated on at the moment. This does not change the fact that Gi coupling plays a role in the observed cAMP reduction of col VI on GPR126.

  12) Several aspects of Fig. 3 are still unclear. I presume the vertical axis of both 3A and 3B refers to vector alone in the absence of forskolin. But, there also appears to be some discrepancy between the two panels Fig. 3A and 3B. In presence of forskolin and the absence of col IV, in panel A, GPR126 is more than 2 X the forskolin treated vector. In Panel B, in the absence of col IV, the cAMP is less than 1.5 X the vector control. This discrepancy is on the same order of magnitude as the col IV effect.

The data shown in Fig. 3A and B are independently performed experiments using transient transfection. These variations are to be expected as transfection efficiency or cell viability naturally vary inbetween assays. When adding forskolin the variability increases even more as the effects are much stronger. This is why we only compare effects within the same assay, which allows us to use a paired t-test for statistical analysis.

13) The data in Fig. 4 has changed from the initial submission. I assume that this is based on repeating the experiments.

That is correct.

14) The paired t-test is only applicable when the samples are actually experimental pairs. Was this the case?

Yes, see comment above.

 15) Something seems to be wrong with the application of the single asterisk to the comparison of the right-hand two bars in Fig. 4. The difference between these two bars seems more significant than any of the other comparisons.

We added the asterix as the statistical analysis indicated it. But if one would want to judge statistical outcome by the looks of the bar graph, then one could see that the error bar is larger in the most right hand bar of Fig. 4A, which already indicates that statistical significance could be smaller.

 16) I do not understand this sentence: “We could previously show that mechanical forces such as shaking and vibration can reverse Gs-inhibition of laminin-211 to Gs-activation that was more efficient than mechanical forces alone [16].”

This sentence refers to our previous paper, where we could show that laminin-211 inhibits Gs-mediated signalling of GPR126. The addition of mechanical forces reverts this inhibition, which results in stronger activation of the receptor compared to the activation that can be achieved through mechanical forces alone.

  17) The manuscript should include a discussion by the authors of their stated reasons from believing that truncation of the protein will not preclude correct folding.

We added the following sentence to the discussion p. 7 ll. 297-300: ”Both ligands bind to a truncated portion of the N terminus, which roughly corresponds to the GAIN domain. As the truncation is designed to start at the endogenous furin site within the N terminus, we believe that this will not preclude correct folding of the protein.” 

Reviewer 2 Report

Authors have now improved the manuscript and addressed many of my concerns satisfactory.

It is however still not clear to me, how the GPR126 T841A mutant would abolish Collagen VI-mediated activation of endogenous GPR126 (ll 234-245) and how that would demonstrate specificity of Collagen VI for GPR126. Do authors rather mean abolishing the effect of heterologously expressed WT GPR126? If so, this sentence needs to be revised.

Also, I do not agree with authors’ argumentation to identify ligand bias: Indeed, in the current system, Collagen VI seems to preferentially stimulate Gi- over Gs-mediated signaling. However, such apparent signaling selectivity may also be due to bias coming from the assay system or other confounding elements (i.e. system bias). Therefore, per definition, ligand bias can only be postulated by using a reference ligand under identical conditions (c.f. Kolb et al., PMID 35106752). Thus, a comparable experiment is needed to hold this up. Alternatively, authors should use a different terminology such as “pathway-preference” instead of “ligand bias”.

Minor:

·      I still could not find a description of the animals used for sciatic nerve preparation.

·      In Figs. 3-4, cAMP levels are given in nM/well. Do authors mean nmol/well or just nM?

Author Response

Authors have now improved the manuscript and addressed many of my concerns satisfactory.

 It is however still not clear to me, how the GPR126 T841A mutant would abolish Collagen VI-mediated activation of endogenous GPR126 (ll 234-245) and how that would demonstrate specificity of Collagen VI for GPR126. Do authors rather mean abolishing the effect of heterologously expressed WT GPR126? If so, this sentence needs to be revised.

The GPR126 T841A mutant exerts a dominant negative effect on the endogenous GPR126. This provides a good indication for receptor specific effects of collagen VI. Knock-out of the receptor either with siRNA or genetically using Crispr Cas9 can also induce dominant negative effects that could mask endogenous responses. As such all these attempts would always just provide a hint. On the other hand, collagen VI can bind to many surface antigens and why would it not activate another molecule? We do show with the mutant that collagen VI reduces cAMP levels only in a functionally expressed GPR126, which would normally preferentially increase cAMP levels through Gs. We do not mean to exclude other molecules but we show that collagen VI acts on GPR126.

Also, I do not agree with authors’ argumentation to identify ligand bias: Indeed, in the current system, Collagen VI seems to preferentially stimulate Gi- over Gs-mediated signaling. However, such apparent signaling selectivity may also be due to bias coming from the assay system or other confounding elements (i.e. system bias). Therefore, per definition, ligand bias can only be postulated by using a reference ligand under identical conditions (c.f. Kolb et al., PMID 35106752). Thus, a comparable experiment is needed to hold this up. Alternatively, authors should use a different terminology such as “pathway-preference” instead of “ligand bias”.

We fully agree with this argument. This signaling bias is clear to us as we always use the tethered peptide as positive control in each assay, where we see cAMP increase in the same assay setup as we see cAMP reduction through collagen VI. We added this control to Fig. 3C. Adding this peptide control it can also be seen that this peptide activates endogenous GPR126, which we have previously demonstrated through siRNA mediated knock-down of GPR126 that abolished this activation (doi:10.1016/j.celrep.2014.11.036). This information is also added to the text p. 6 ll. 243-244.

 Minor:

  • I still could not find a description of the animals used for sciatic nerve preparation.

We used adult (postnatal day 45-60) C57/BL6 mouse sciatic nerves.

  • In Figs. 3-4, cAMP levels are given in nM/well. Do authors mean nmol/well or just nM?

Thank you for pointing this out! We changed it accordingly.

Round 3

Reviewer 1 Report

    The current version of the manuscript is significantly improved. The addition of the supplementary methodological information on immunoprecipitation and mass spectrometry and the indicated revisions of the text are welcome modifications. I have only the following remaining minor issues:

1) The information about false discovery rates in the authors’ response to the previous point #2 seems contradictory: “the data was filtered to between a one and two percent peptide false discovery rate.  A total of four independent experiments were performed. The false discovery rates are 0.48%, 0.12%, 0.65% and 0.28%.” Usually, false discovery rate is set as a software parameter.

 2) The authors’ response to the previous point #5 about recovering only three peptides from collagen IV is that many ECM proteins are crosslinked to other ECM proteins and are difficult to detect. Since detection is of proteolytic peptides, it is not clear why crosslinking should be a problem. If this is, in fact a problem, it raises the possibility that other (undetected) WCM proteins may interact with GPR126, which should be acknowledged in the text.

 3) In point #8 of my previous review, I should probably have used the term “input” instead of lading control. In co-immunoprecipitation, it is customary to show a direct immunoblot of the material used for the precipitation of different samples to give an idea of how the amount of immunoprecipitated material (collagen VI) compares to the amount of material used as input for the co-IP. This could also show that differences between samples are the direct result of differences in pull-down and not differences in expression of the target or sample manipulation, though this is apparently not relevant here, since all IPs were presumably from the same sample.  

Author Response

The current version of the manuscript is significantly improved. The addition of the supplementary methodological information on immunoprecipitation and mass spectrometry and the indicated revisions of the text are welcome modifications. I have only the following remaining minor issues:

1) The information about false discovery rates in the authors’ response to the previous point #2 seems contradictory: “the data was filtered to between a one and two percent peptide false discovery rate.  A total of four independent experiments were performed. The false discovery rates are 0.48%, 0.12%, 0.65% and 0.28%.” Usually, false discovery rate is set as a software parameter.

Response: We apologize for our miswording. We have since removed the sentence “the data was filtered to between a one and two percent peptide false discovery rate.”  

 2) The authors’ response to the previous point #5 about recovering only three peptides from collagen IV is that many ECM proteins are crosslinked to other ECM proteins and are difficult to detect. Since detection is of proteolytic peptides, it is not clear why crosslinking should be a problem. If this is, in fact a problem, it raises the possibility that other (undetected) WCM proteins may interact with GPR126, which should be acknowledged in the text.

Response: We used the nerve homogenates as input for the pull-down assay without protein digestion. Proteolytic peptides were generated after the GPR126 NTF pull-down experiment. Therefore, the crosslinking status of the ECM significantly impede one’s ability in ligand binding assay. This is one the major hurdle in ECM biology.

 3) In point #8 of my previous review, I should probably have used the term “input” instead of lading control. In co-immunoprecipitation, it is customary to show a direct immunoblot of the material used for the precipitation of different samples to give an idea of how the amount of immunoprecipitated material (collagen VI) compares to the amount of material used as input for the co-IP. This could also show that differences between samples are the direct result of differences in pull-down and not differences in expression of the target or sample manipulation, though this is apparently not relevant here, since all IPs were presumably from the same sample.

Response: We thank the reviewer #1 on the statement “…though this is apparently not relevant here, since all IPs were presumably from the same sample.”

Reviewer 2 Report

Authors have answered all my concerns satisfactory. Just a few minor issues remain:

- The newly added peptide control (pGPR126) in Fig 3C is not mentioned in the caption and methods (please add concentrations).

- PTX as an abbreviation is used twice for Pertussis toxin and Pentraxin

Author Response

We want to express our sincere gratefulness to the reviewer for the repeated time invested into reading our manuscript. All remarks were truly helpful and majorly improved the story.

Authors have answered all my concerns satisfactory. Just a few minor issues remain:

- The newly added peptide control (pGPR126) in Fig 3C is not mentioned in the caption and methods (please add concentrations).

We added a statement to the methods section concerning peptide synthesis, dilution and concentration at p. 2, ll. 81-87 and p.3 ll. 179-180. Also, the caption of Fig. 3 mentions this now.

- PTX as an abbreviation is used twice for Pertussis toxin and Pentraxin

Thank you very much for detecting this double use of abbreviation! We used now the abbreviation (Pent) for the Pentraxin domain and changed this in text at all mentionings and in the Figure 1A.